# Short- and Long-Term Survival among Elderly Colorectal Cancer Patients in Finland, 2006–2015: A Nationwide Population-Based Registry Study

**DOI:** 10.3390/cancers16010135

**Published:** 2023-12-27

**Authors:** Tanja Hukkinen, Tobias Olenius, Selja Koskensalo, Anna Lepistö, Laura Koskenvuo, Camilla Böckelman

**Affiliations:** Department of Gastroenterological Surgery, Abdominal Center, Helsinki University Hospital, Faculty of Medicine, University of Helsinki, Meilahden Sairaalakampus, Palvelukeskus PAKE.PA3.30, Stenbäckinkatu 9A, PL 440, 00029 Helsinki, Finland; tanja.hukkinen@helsinki.fi (T.H.);

**Keywords:** colorectal cancer, right-sided colon cancer, left-sided colon cancer, rectal cancer, elderly, short-term survival, long-term survival, OS, DSS, Finland

## Abstract

**Simple Summary:**

This study aimed to assess the short- and long-term survival of elderly (≥75 years old) CRC patients. Survival was analyzed according to tumor location, cancer stage, and age group. Our results showed that CRC survival among elderly patients with localized or locally advanced disease is generally good when considering the age of patients. In particular, the 75–79 and 80–84 age groups exhibited fairly good survival when compared with younger age groups. The long-term overall survival of patients aged ≥ 85 was, as expected, worse than in younger patients. However, the postoperative short-term survival for patients eligible for surgery was good, taking into account that we also included emergency procedures. These findings emphasize the importance of optimizing care for elderly CRC patients.

**Abstract:**

This population-based registry study aimed to report 30-day and one-year postoperative survival, five-year overall survival (OS), and disease-specific survival (DSS) among elderly (≥75 years old) colorectal cancer (CRC) patients. All new colorectal cancer cases from 2006–2015 were included and followed until death or the end of follow-up (end of 2016). Among 27,088 CRC patients, 11,306 patients were ≥75 years old. Among patients ≥ 75 years, 36.8% (*n* = 4160) had right-sided colon cancer, 21.9% (*n* = 2478) left-sided colon cancer, and 32.3% (*n* = 3650) rectal cancer. In this study population, 932 patients were aged ≥ 90. The 30-day postoperative OS for patients aged 75–79 was 96.1% (95% confidence interval [CI] 95.3–96.9) falling to 93.2% (95% CI 92.0–94.4) for patients aged 80–84. The one-year postoperative OS among patients aged 75–79 was 86.3% (95% CI 84.7–87.9) compared with 80.5% (95% CI 78.7–82.3) among patients aged 80–84. Five-year OS among patients aged 75–79 was 47.6% (95% CI 46.0–49.2) and 36.6% (95% CI 34.8–38.4) among patients aged 80–84, compared with 61.7% (95% CI 60.9–62.5) among younger patients (<75 years old). Survival among elderly CRC patients (≥75 years old) was in general fairly good when compared with younger patients, especially among patients aged 75–79 and 80–84 with localized or locally advanced disease.

## 1. Introduction

As life expectancy in the Nordic countries increases [1], we face an increasingly aging population. In Finland, colorectal cancer (CRC) is the second most common cancer, with 3685 new cases reported in 2021 [2]. The incidence of CRC increases with age, and as the proportion of elderly in the population has increased, so, too, has the incidence of CRC in the population [3].

The existing excess mortality among elderly CRC patients primarily stems from death due to causes other than CRC, as found in a Dutch study consisting of 9397 patients [4]. When survival data were corrected for expected death from other causes, no differences between age groups emerged. According to a registry study in the United States among 24,426 colon cancer patients, an older age associated with a lower proportion of colon cancer deaths and a higher proportion of deaths due to cardiovascular disease [5].

Among elderly CRC patients undergoing surgery, the first year of recovery is particularly important given the high morbidity observed [6]. However, after surviving the first year postoperatively, survival is comparable to that among younger age groups [7]. Morbidity postoperatively can be minimized through the evaluation and optimization of patients, especially among older patients [8].

Yet, only a few previous population-based studies have examined CRC stage-dependent survival among elderly patients. In the past, research into survival among older CRC patients has, to our knowledge, focused more on general survival and not on stage-specific survival. Previous studies have also primarily examined right-sided versus left-sided colon cancer [9]. With populations growing older, further exploring survival at different stages of CRC is of interest, specifically in elderly populations. Thus, this nationwide population-based registry study aimed to explore elderly CRC patients’ short- and long-term overall survival (OS) based on tumor spread, tumor location, and patient age groups, as well as disease-specific survival (DSS).

## 2. Materials and Methods

### 2.1. Patients

This study comprised all Finnish CRC patients diagnosed between 2006 and 2015. We received data from the Finnish Cancer Registry (FCR), which included patients’ gender and age at diagnosis, tumor-specific data (stage and location), and cause and time of death data. The completeness of the data for CRC was 97.4% [10], with the accuracy of the CRC stage and location data considered good [11]. Patients were divided into age groups as follows: <75 years old, 75–79, 80–84, 85–89, and ≥90. Different age groups were then compared, with the <75 age group serving as the control group.

Altogether, there were 27,088 CRC cases. Among patients aged <75 years, 9035 (57.2%) were male. Among patients aged ≥ 75, 5194 (45.9%) were male. For patients with secondary or tertiary cancer, a case was only included when the patient was diagnosed with cancer for the first time. We excluded patients with appendix cancer and patients with missing information. One patient had a missing date of death and was thus excluded from all analyses.

### 2.2. Variables

We received information on tumor stages based on FCR’s classification: 0, unknown; 1, localized; 2, non-localized, regional lymph node metastasis only; 3, metastasized further than to regional lymph nodes or invading adjacent tissues; 4, non-localized, no information on extent; 5, locally advanced, tumor invasion to adjacent tissues; and 6, non-localized, including distant lymph node metastasis. The FCR classification is, to a large extent, comparable with the TNM staging (Appendix A [11]).

Results were analyzed according to tumor location: right colon, left colon, and rectum, where tumors were considered right sided if they were located between the caecum and the end of the transverse colon. Tumors were considered left sided if they were located within the area from the splenic flexure of the colon to the distal end of the rectosigmoid junction. Patients with an unknown tumor location were labeled “other” in patient characteristics and excluded from the tumor location analyses (*n =* 2918; Table 1).

Surgical data were received for 48.3% of patients. We observed 30-day and one-year postoperative OS. Patients undergoing any resection were included in the postoperative analyses. Patients with metastatic disease (FCR 6) were excluded from these analyses.

### 2.3. Statistical Analysis

We reported the median and interquartile range (IQR) for ages at diagnosis. We estimated DSS and OS using the Kaplan–Meier method and log-rank analyses to calculate the *p* value. We considered *p* < 0.05 as statistically significant. DSS was calculated from the date of diagnosis until death due to CRC or until the end of follow-up (end of 2016). OS was calculated from the date of diagnosis until death. Postoperative OS was calculated from the date of surgery until death or until the end of follow-up. The risk of death was calculated using hazard ratios (HRs) comparing older patients with patients ≥ 75 years using a univariate Cox regression analysis. We reported the 95% confidence intervals (CIs) for five-year survival and compared CIs between groups. Patients with missing operative variables were excluded from the postoperative survival analyses only. All statistical analyses were calculated using SPSS Statistics, version 26 (IBM, Armonk, NY, USA).

### 2.4. Permissions

The study protocol was approved by the Finnish Institute of Health and Welfare (THL/722/5.05.00/2018, extension THL/3562/14.06.00/2022).

## 3. Results

Among 27,088 CRC patients, 11,306 were ≥75 years old. The median age at diagnosis among elderly patients ≥ 75 was 81.8 years (IQR 78.3–85.8). Surgical information was available for 13,076 patients, of whom 5460 were ≥75 years. Resections were performed on 5146 patients ≥ 75 years. Among patients ≥ 75 years, 36.8% had right-sided colon cancer, 21.9% left-sided colon cancer, and 32.3% rectal cancer. In this study population, 932 patients were ≥90 years (Table 1). Five-year OS for all CRC patients diagnosed in Finland between 2006 and 2015 was 51.0% (95% CI 50.4–51.6). For patients < 75 years, five-year OS was 61.7% (95% CI 60.9–62.5), falling to 36.0% (95% CI 35.0–37.0) for elderly patients ≥ 75 years. For patients ≥ 90 years, the five-year OS was 9% (95% CI 6.8–11.2) and the median OS was 0.50 years (95% CI 0.40–0.59).

### 3.1. Five-Year Overall Survival Based on FCR Group

Patients with localized disease (FCR 1) aged 75–79 had a five-year OS of 72.7% (95% CI 69.4–76.0), falling to 54.6% (95% CI 50.5–58.7) among those aged 80–84. Among patients with locally advanced disease where the tumor invaded adjacent tissues (FCR 5), the OS for patients aged 75–79 was 60.4% (95% CI 55.1–65.7) and 52.9% (95% CI 47.2–58.6) for patients aged 80–84. Among patients with regional lymph node metastasis (FCR 2) aged 75–79, OS was 52.2% (95% CI 47.3–57.1), falling to 36.8% (95% CI 31.7–41.9) for those aged 80–84. Patients with metastatic disease (FCR 6) aged 75–79 and 80–84, respectively, had a five-year OS of just over 4% (Table 2 and Figure 1).

Among all CRC patients aged 75–79 in the study population, OS was slightly worse when compared with patients < 75 years (HR 1.62, 95% CI 1.54–1.69; Table 3). Among patients with localized disease (FCR 1), better survival was found for patients aged 75–79 (HR 2.83, 95% CI 2.43–3.29) compared with patients aged 80–84 (HR 5.10, 95% CI 4.43–5.89). We observed similar survival for patients aged 75–79 and 80–84, with locally advanced disease where the tumor invaded adjacent tissues (FCR 5) (HR 2.04, 95% CI 1.73–2.40 vs. HR 2.99, 95% CI 2.54–3.51; Table 3).

### 3.2. Five-Year Overall Survival Based on Tumor Location

In the 75–79 age group, five-year OS was 49.2% (95% CI 46.1–52.1) for right-sided disease, 47.6% (95% CI 44.3–50.9) for left-sided disease, and 46.7% (95% CI 44.0–49.4) for rectal cancer (Table 4).

Patients with right-sided colon cancer with localized disease (FCR 1) aged 75–79 had a five-year OS of 73.3% (95% CI 67.2–79.4), falling to 55.4% (95% CI 48.5–62.3) among patients aged 80–84. The five-year OS for patients aged 75–79 with locally advanced disease in which the tumor invaded adjacent tissues (FCR 5) was 64.2% (95% CI 55.6–72.8) and 59.2% (95% CI 51.2–67.2) among patients aged 80–84. Patients with regional lymph node metastasis (FCR 2) aged 75–79 had a five-year OS of 49.9% (95% CI 42.3–57.5) compared with 37.3% (95% CI 29.3–45.3) among those aged 80–84 (Table 4A and Figure 2).

The five-year OS for patients with left-sided colon cancer with localized disease (FCR 1) aged 75–79 was 73.4% (95% CI 66.3–80.5), falling to 51.7% (95% CI 42.7–60.7) among patients aged 80–84. For patients with locally advanced left-sided colon cancer invading to an adjacent structure, but without lymph node or distant metastasis (FCR 5) aged 75–79, OS was 60.4% (95% CI 50.8–70.0), falling to 47.0% (95% CI 35.0–59.0) among patients aged 80–84. Patients with left-sided colon cancer with regional lymph node metastasis (FCR 2) aged 75–79 had a five-year OS of 53.0% (95% CI 43.6–62.4) compared with 37.8% (95% CI 27.4–48.2) among patients aged 80–84 (Table 4B and Figure 3).

Finally, rectal cancer patients with localized disease (FCR 1) aged 75–79 had a five-year OS of 70.3% (95% CI 65.0–75.6) compared with an OS of 55.9% (95% CI 49.0–62.8) among those aged 80–84. For rectal cancer patients with locally advanced disease invading to an adjacent structure but without lymph node or distant metastasis (FCR 5), the five-year OS was 55.0% (95% CI 45.6–64.4) among patients aged 75–79 compared with 47.6% (95% CI 37.0–58.2) among those aged 80–84. Patients with regional lymph node metastasis (FCR 2) aged 75–79 had a five-year OS of 54.1% (95% CI 45.9–62.3), falling to 35.9% (95% CI 27.1–44.7) among those aged 80–84 (Table 4C and Figure 4).

### 3.3. Thirty-Day and One-Year Postoperative Overall Survival

Patients with localized disease (FCR 1) aged 75–79 had a 30-day postoperative OS of 97.2% (95% CI 95.6–98.8), falling to 93.4% (95% CI 90.7–96.1) among those aged 80–84. Among patients with locally advanced disease in which the tumor invaded adjacent tissues (FCR 5) aged 75–79, the 30-day postoperative OS was 96.9% (95% CI 95.1–98.7) compared with 95.7% (95% CI 93.5–97.9) among patients aged 80–84. Among patients with regional lymph node metastasis (FCR 2), the 30-day postoperative OS was 97.5% (95% CI 95.9–99.1) among patients aged 75–80, falling slightly to 94.5% (95% CI 92.0–97.0) for patients aged 80–84 (Table 5).

The one-year postoperative OS for patients with localized disease (FCR 1) aged 75–79 was 92.9% (95% CI 90.4–95.4) and 88.0% (95% CI 84.5–91.5) among patients aged 80–84. Among patients with locally advanced disease in which the tumor invaded adjacent tissues (FCR 5), one-year postoperative OS was 92.6% (95% CI 89.9–95.3) for patients aged 75–79, falling to 86.5% (95% CI 82.8–90.2) among patients aged 80–84. Patients aged 75–79 with regional lymph node metastasis (FCR 2) had a one-year postoperative survival of 89.3% (95% CI 86.2–92.4), falling to 81.4% (95% CI 76.9–85.9) among patients aged 80–84 (Table 6).

### 3.4. Five-Year Disease-Specific Survival Based on FCR Groups

Patients with localized disease (FCR 1) aged 75–79 had a five-year DSS of 90.6% (95% CI 88.2–93.0) compared with 81.2% (95% CI 77.7–84.7) among patients aged 80–84 (Table 4). For patients with locally advanced disease in which the tumor invaded adjacent tissues (FCR 5), the five-year DSS was 75.5% (95% CI 70.8–80.2) among patients aged 75–79 and 73.0% (95% CI 67.7–78.3) among those aged 80–84. DSS for patients with regional lymph node metastasis (FCR 2) aged 75–79 was 61.6% (95% CI 56.7–66.5) and 50.6% (95% CI 45.1–56.1) among patients aged 80–84. All patients ≥ 75 with metastatic disease (FCR 6) had a five-year DSS of around 5% compared with 17.8% (95% CI 15.6–20.0) among patients < 75 years (Appendix A).

### 3.5. Five-Year Disease-Specific Survival among FCR Groups Based on Tumor Location

In the 75–79 age group, five-year DSS was 61.1% (95% CI 58.4–63.8) for right-sided disease, 60.0% (95% CI 56.7–63.3) for left-sided disease, and 57.2% (95% CI 54.3–60.1) for rectal cancer (Appendix A).

Patients with right-sided colon cancer with localized disease (FCR 1) aged 75–79 had a five-year DSS of 92.7% (95% CI 89.0–96.4), falling to 83.1% (95% CI 77.8–88.4) among patients aged 80–84. The five-year DSS for patients with locally advanced disease where the tumor invaded adjacent tissues (FCR 5) was 83.5% (95% CI 76.8–90.2) among patients aged 75–79 and 83.2% (95% CI 76.7–89.7) among patients aged 80–84. Patients with regional lymph node metastasis (FCR 2) had a five-year DSS of 57.2% (95% CI 49.4–65.0) among patients aged 75–79 compared with 55.7% (95% CI 47.3–64.1) among those aged 80–84 (Appendix A).

The five-year DSS for patients with left-sided colon cancer with localized disease (FCR 1) was 94.0% (95% CI 90.1–97.9) among those aged 75–79, falling to 73.9% (95% CI 65.7–82.1) among patients aged 80–84. For patients with locally advanced left-sided colon cancer invading to an adjacent structure but without lymph node or distant metastasis (FCR 5) aged 75–79, DSS was 76.2% (95% CI 67.6–84.8), falling to 65.3% (95% CI 54.7–75.9) among patients aged 80–84. Patients with left-sided colon cancer with regional lymph node metastasis (FCR 2) aged 75–79 had a five-year DSS of 66.3% (95% CI 57.1–75.5) compared with 51.6% (95% CI 40.4–62.8) among patients aged 80–84 (Appendix A).

Finally, rectal cancer patients with localized disease (FCR 1) aged 75–79 had a five-year DSS of 86.6% (95% CI 82.3–90.9) compared with a DSS of 82.9% (95% CI 77.4–88.4) among those aged 80–84. For rectal cancer patients with locally advanced disease invading to an adjacent structure but without lymph node or distant metastasis (FCR 5), the five-year DSS was 66.4% (95% CI 57.0–75.8) among patients aged 75–79 compared with 63.6% (95% CI 52.0–75.2) among those aged 80–84. Patients with regional lymph node metastasis (FCR 2) aged 75–79 had a five-year DSS of 62.3% (95% CI 53.9–70.7), falling to 46.3% (95% CI 36.7–55.9) among those aged 80–84 (Appendix A).

## 4. Discussion

This study, consisting of 11,306 Finnish CRC patients ≥ 75 years old diagnosed between 2006 and 2015, shows that the five-year OS and DSS among elderly patients with CRC is fairly good. In particular, patients aged 75–79 and 80–84 with localized and locally advanced disease experienced a relatively good long-term OS. Furthermore, we observed a similar DSS among patients < 75 years and ≥75 years, particularly among patients with localized disease.

Our results demonstrate that the five-year OS among elderly patients is clearly worse compared with the five-year DSS. Patients with localized disease aged 75–79 had an OS exceeding 70% and those aged 80–84 had an OS of over 50%, whereas the respective DSS is remarkably better (over 90% and 80%). A similar pattern among elderly CRC patients was noted in a recent study based on the SEER database [12]. That study demonstrated that elderly patients undergoing surgery usually have a better OS and DSS than those receiving palliative treatment alone. This trend was also observed in a recent study among 1482 patients from 2005–2020, which revealed a larger influence of age on OS compared with cancer-specific survival [13]. These findings emphasize the influence of age on OS and the importance of taking OS into account in treatment decisions. Given that almost half of the elderly patients with localized disease passed away within five years of diagnosis (five-year OS 56.7%, 95% CI 54.3–59.1), treatment decisions must acknowledge that these patients may have a relatively short life expectancy despite surviving CRC.

Contrary to young patients, OS among elderly patients appeared worse in patients with more distal gastrointestinal tumors (Table 4). According to our results, elderly patients with right-sided colon cancer in general exhibited a better long-term OS than those with left-sided colon cancer or rectal cancer. Our findings differ from earlier studies, suggesting that elderly patients with left-sided colon cancer have a better OS than patients with right-sided disease [14,15,16]. Yet, such reports have included younger patients, which likely affected the results since younger patients usually present with more distal disease [17]. A recent study among 91,416 patients suggested that those with right-sided colon cancer exhibited a worse OS than patients with left-sided colon cancer; yet, after propensity score matching, the results agreed with our findings, since survival (both overall and disease-specific) was better in patients with right-sided colon cancer compared with left-sided disease [18]. Elderly CRC patients in our dataset were also more often female. Earlier studies reported a lower mortality and incidence of CRC among women than among men in younger age groups [19,20]. Based on these results, it seems that CRC in women emerges at a later age. In our study, the differences in OS between right-sided and left-sided colon cancer became more obvious among patients aged 80–84 and older, after which right-sided colon cancer patients exhibited better survival.

Rectal cancer patients with locally advanced disease had, according to a recent retrospective study from 2004–2018 among 328 patients, a similar DSS to that among younger age groups [21]. These results differ from ours, since elderly patients with locally advanced disease in which the tumor invaded adjacent tissues (FCR 5) had a worse five-year DSS and OS than that observed among younger patients. Although survival is worse than that among younger patients, the five-year DSS continues to exceed 60% for all age groups. Our patients with locally advanced disease had an OS over 40% up to age 89. The poorer survival among rectal cancer patients may be explained by aggressive rectal cancer care, including extensive surgeries, which are taxing to the elderly already burdened by comorbidities [22,23]. The number of emergency surgeries among older CRC patients remains higher than that among younger patients, as demonstrated by a population-based study [24]. The authors of that study suggested that because older patients more rarely undergo surgery and more often present with tumor-related bowel obstruction, they, therefore, more often require emergency surgery. This is supported by another national cohort study among 31,665 patients, in which 17% of patients > 75 years old presented with bowel obstructions at diagnosis compared with 11% of younger patients [25]. In the same study, patients > 75 years old received adjuvant (15% vs. 29%) and palliative (48% vs. 85%) chemotherapy less often than younger patients. A large proportion (40.1%) of patients ≥ 90 in our dataset were classified as FCR 0, unknown stage. Given that the FCR has a completeness of data reaching 97.4% [10], some of the cases with incomplete data need to be categorized as FCR 0. Hence, the high proportion of FCR 0 among patients ≥ 90 may be explained by such patients receiving palliative treatment upon cancer detection. Furthermore, some cancers may be registered only upon postmortem examination.

A Spanish retrospective study found that CRC patients > 75 years remain undertreated [26]. Specifically, elderly patients in the study were less likely to undergo surgery and receive chemotherapy, subsequently resulting in lower DSS rates. In this light, our results appear important to highlight, since they offer a solid overview of the situation in Finland, where elderly CRC patients are predominantly treated surgically in the same way as younger patients. Compared with 30-day postoperative OS, we observed a one-year postoperative OS that is significantly worse among patients ≥ 80. The poorer postoperative OS among the older age groups likely results from death due to other causes and the influence of comorbidities. Moreover, we included emergency surgeries in our study. The mortality of the elderly in emergency settings is worse than that in younger patients, which must be taken into account [27]. The median postoperative hospital stay for elderly patients in our dataset was similar when compared with that in younger patients. The one-year postoperative OS of 62.9–86.3% (average 76.2%) among patients ≥ 75 years old agrees with a retrospective database study of 232 patients, where the one-year postoperative survival was 73.5% among patients > 80 years [28]. Taken together with the findings from our study concerning overall survival, which included the postoperative hospital stay, this highlights the importance of widely applying standardized treatment and the geriatric assessment of older patients. In a recent population-based Finnish study, risk factors for severe postoperative complications were analyzed, shedding light on various comorbidities to consider when assessing elderly patients preoperatively [29]. The elderly CRC population is a quite heterogeneous group with multiple comorbidities, although this alone should not serve as the basis for not offering surgery with a curative intent [30]. As such, the Clinical Frailty Scale (CFS) is a tool for assessing preoperative cognitive and physical independence and activity, recently validated in a Finnish multicenter study [8]. Furthermore, patients with a lower CFS score were less likely to develop postoperative complications. Previous studies demonstrated the prognostic value of the CFS for elderly patients’ survival [31,32]. Thus, frailty and comorbidities are most likely more important factors to evaluate as opposed to age alone when assessing treatment options [33,34].

This study has several strengths and limitations. The data source for this study is exceptionally comprehensive and reliable since all healthcare organizations in Finland are obligated by law to report information about diagnosed cancer cases to the Finnish Cancer Registry (FCR). Cause of death data came from Statistics Finland, a reliable source for the cause and date of all deaths in the country. The accuracy of tumor characteristics recorded in FCR is good, with a completeness of data for CRC reaching 97.4% [10]. The registration of a cancer case is linked to a patient’s unique social security number, making it possible to link patient information across different registries. When a cancer case is reported, the healthcare organization must report the ICD-10 code, the time of diagnosis, a pathologic–anatomical diagnosis, and cancer staging information [35].

One limitation of this study is that with increasing age, patient groups become smaller. In addition, since our material is registry based, we lacked information on adjuvant treatment and recurrence. More specifically, we received no information on surgery for 51.7% of patients. This may in part be due to the fact that the elderly are more often directed to palliative care, at times due to a lack of information. Thus, the results on postoperative survival should be interpreted with caution. Furthermore, the classification system used by FCR differs from the UICC TNM staging, although the FCR classification is rather thorough, making it possible to categorize many cancer cases. Yet, the specificity of the FCR classification might make comparisons across clinical studies difficult, although it provides exceptional detail on the tumor characteristics and recognizes similar groupings as the TNM stages [11]. Another strength of this study is its large nationwide cohort of CRC patients, consisting of 27,088 CRC patients, from whom 11,306 were ≥75 years old. As a result, this study sheds light on the outcomes for older CRC patients on a national level from a country with a high standard of health care offered by multiprofessional teams.

## 5. Conclusions

This study demonstrates that Finnish colorectal cancer patients ≥ 75 years old experience a fairly good OS. The long-term OS of patients aged ≥ 85 was, as expected, notably worse than in younger age groups. However, the postoperative short-term survival among patients eligible for surgery, taking into account that we also included emergency procedures, was good since 85–89-year-old patients experienced a 30-day postoperative survival of >90%.

## Figures and Tables

**Figure 1 cancers-16-00135-f001:**
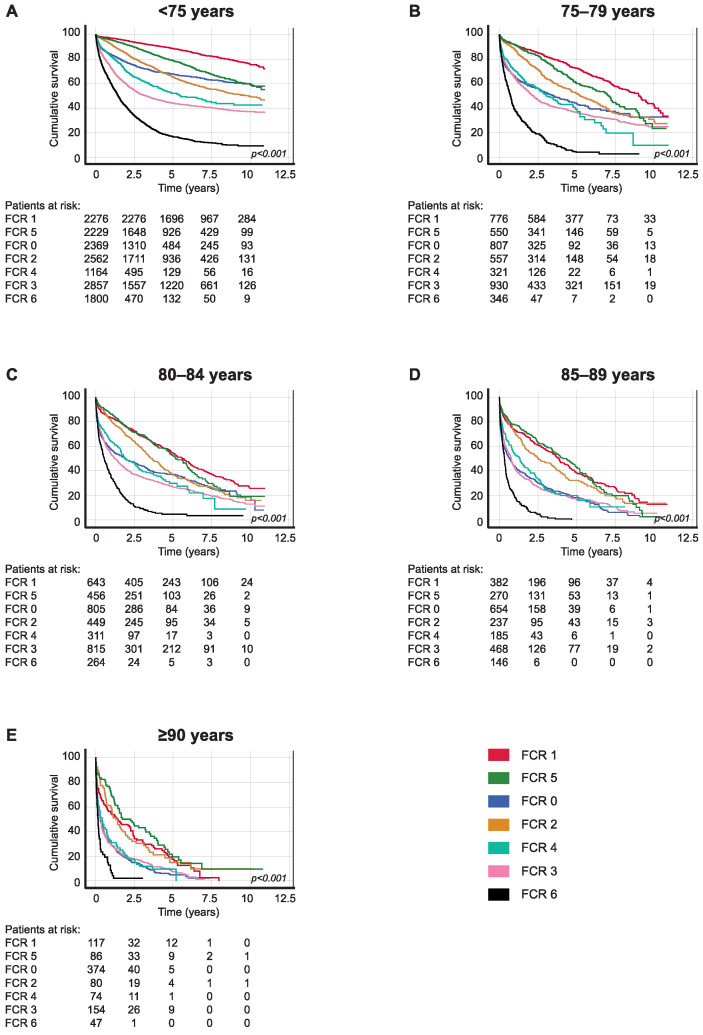
Overall survival analysis for colorectal cancer patients diagnosed in 2006–2015 according to age at diagnosis: (**A**) <75, (**B**) 75–79, (**C**) 80–84, (**D**) 85–89, and (**E**) ≥90. Finnish Cancer Registry classes: 0, unknown; 1, localized; 2, non-localized, regional lymph node metastasis only; 3, metastasized further than to regional lymph nodes or invading adjacent tissues; 4, non-localized, no information on extent; 5, locally advanced, tumor invasion to adjacent tissues; and 6, non-localized, including distant lymph node metastasis. The *p*-value calculated using the log-rank test.

**Figure 2 cancers-16-00135-f002:**
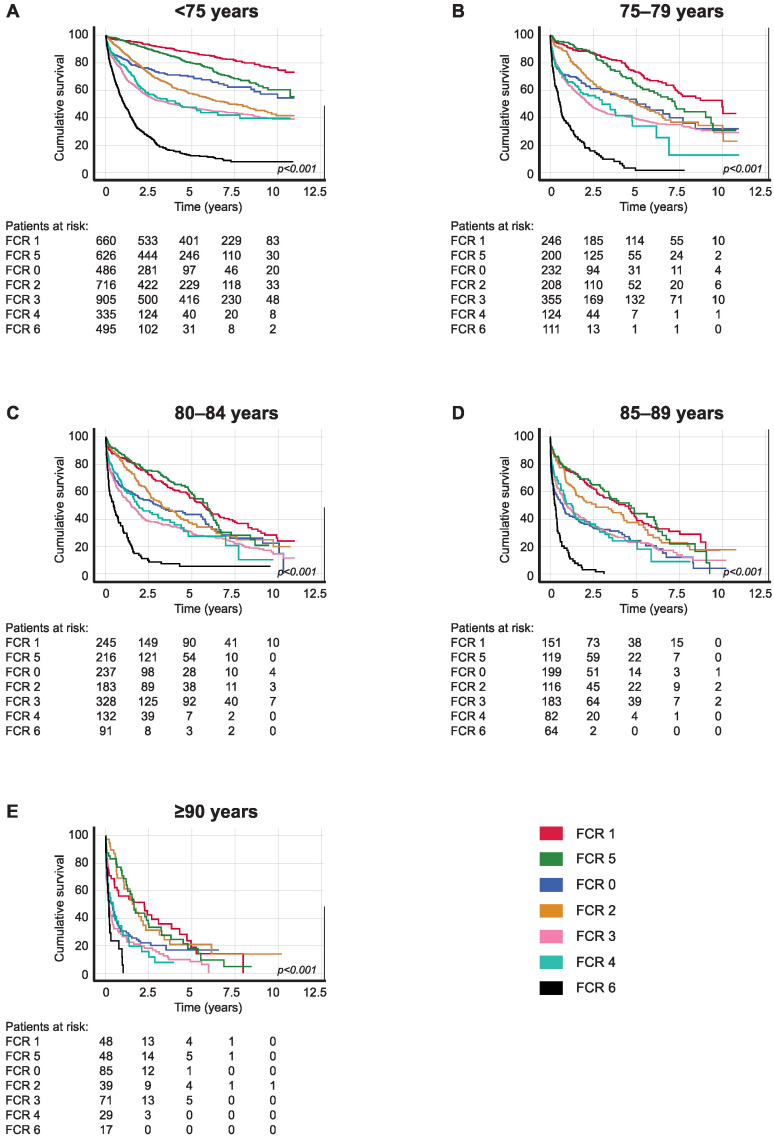
Overall survival analysis for right-sided colon cancer patients diagnosed in 2006–2015 according to age at diagnosis: (**A**) <75, (**B**) 75–79, (**C**) 80–84, (**D**) 85–89, and (**E**) ≥90. Finnish Cancer Registry classes: 0, unknown; 1, localized; 2, non-localized, regional lymph node metastasis only; 3, metastasized further than to regional lymph nodes or invading adjacent tissues; 4, non-localized, no information on extent; 5, locally advanced, tumor invasion to adjacent tissues; and 6, non-localized, including distant lymph node metastasis. The *p*-value calculated using the log-rank test.

**Figure 3 cancers-16-00135-f003:**
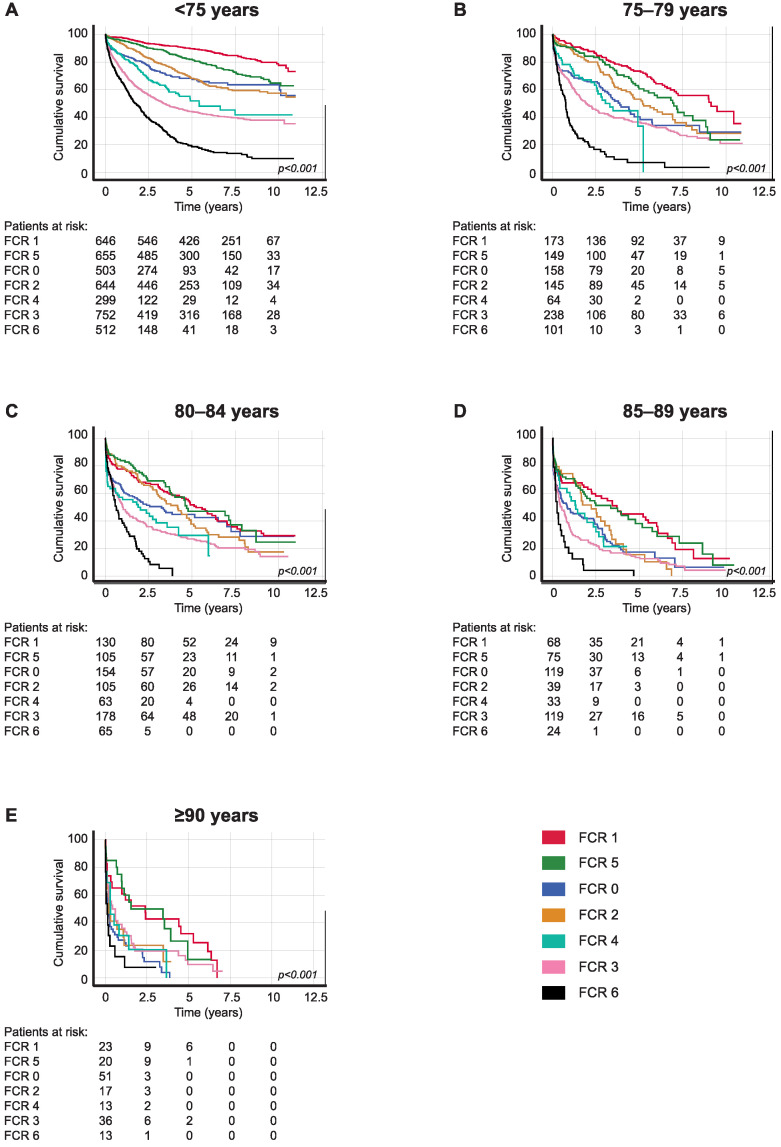
Overall survival analysis for left-sided colon cancer patients diagnosed in 2006–2015 according to age at diagnosis: (**A**) <75, (**B**) 75–79, (**C**) 80–84, (**D**) 85–89, and (**E**) ≥90. Finnish Cancer Registry classes: 0, unknown; 1, localized; 2, non-localized, regional lymph node metastasis only; 3, metastasized further than to regional lymph nodes or invading adjacent tissues; 4, non-localized, no information on extent; 5, locally advanced, tumor invasion to adjacent tissues; and 6, non-localized, including distant lymph node metastasis. The *p*-value calculated using the log-rank test.

**Figure 4 cancers-16-00135-f004:**
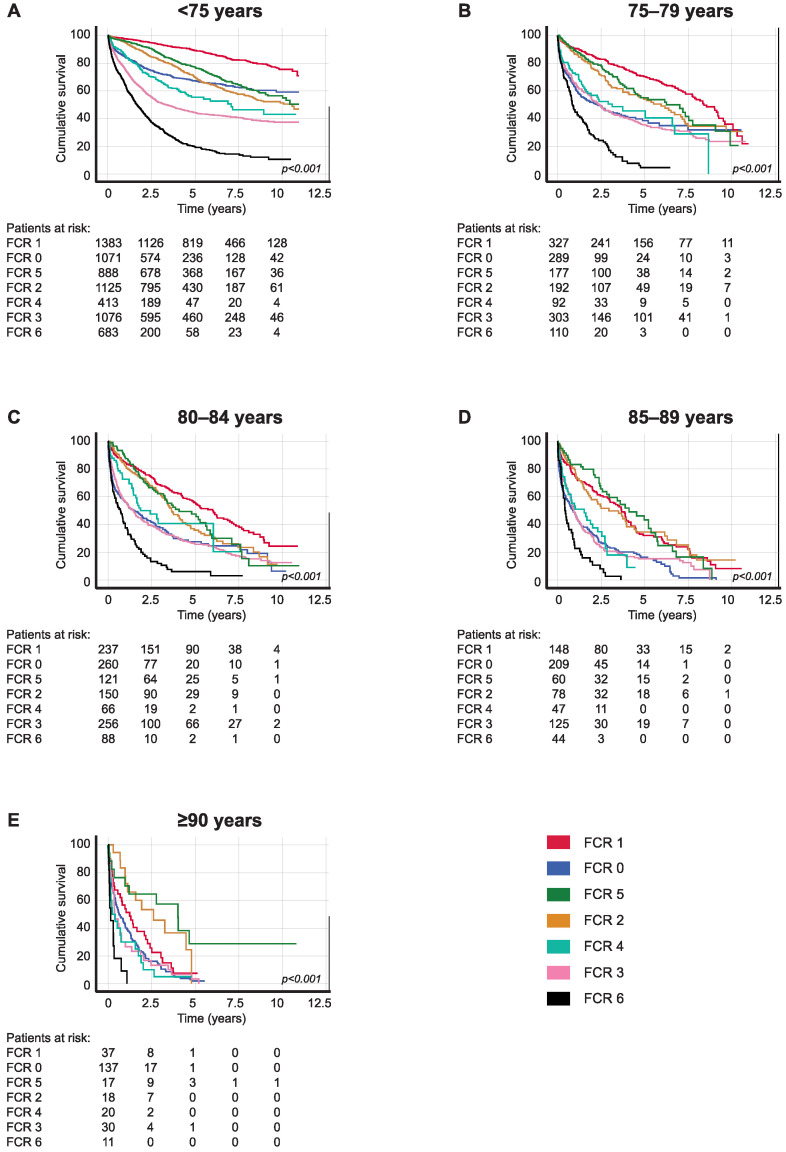
Overall survival analysis for rectal cancer patients diagnosed in 2006–2015 according to age at diagnosis: (**A**) <75, (**B**) 75–79, (**C**) 80–84, (**D**) 85–89, and (**E**) ≥90. Finnish Cancer Registry classes: 0, unknown; 1, localized; 2, non-localized, regional lymph node metastasis only; 3, metastasized further than to regional lymph nodes or invading adjacent tissues; 4, non-localized, no information on extent; 5, locally advanced, tumor invasion to adjacent tissues; and 6, non-localized, including distant lymph node metastasis. The *p*-value calculated using the log-rank test.

**Table 1 cancers-16-00135-t001:** Patient characteristics for 27,088 colorectal cancer patients treated in Finland, 2006–2015.

Patient Characteristics						
	**<75 Years Old**	**≥75 Years Old**	**75–79 Years Old**	**80–84 Years Old**	**85–89 Years Old**	**≥90 Years Old**
	***n*** **(%)**	***n*** **(%)**	***n*** **(%)**	***n*** **(%)**	***n*** **(%)**	***n*** **(%)**
**All**	15,782	11,306	4287	3745	2342	932
**Gender**						
Male	9035 (57.2)	5194 (45.9)	2302 (53.7)	1732 (46.2)	877 (37.4)	283 (30.4)
Female	6747 (42.8)	6112 (54.1)	1985 (46.3)	2013 (53.8)	1465 (62.6)	649 (69.6)
**FCR classification ^1^**						
0	2375 (15.1)	2641 (23.4)	807 (18.8)	803 (21.5)	654 (27.9)	374 (40.1)
1	2782 (17.6)	1919 (17.0)	776 (18.1)	644 (17.2)	382 (16.3)	117 (12.6)
2	2565 (16.3)	1321 (11.7)	557 (13.0)	449 (12.0)	237 (10.1)	80 (8.6)
3	2859 (18.1)	2367 (20.9)	930 (21.7)	815 (21.8)	468 (20.0)	154 (16.5)
4	1166 (7.4)	891 (7.9)	321 (7.5)	311 (8.3)	185 (7.9)	74 (7.9)
5	2233 (14.1)	1362 (12.0)	550 (12.8)	456 (12.2)	270 (11.5)	86 (9.2)
6	1802 (11.4)	803 (7.1)	346 (8.1)	264 (7.0)	146 (6.3)	47 (5.1)
**Tumor location**						
Right colon	4228 (26.8)	4160 (36.8)	1476 (34.4)	1433 (38.3)	914 (39.0)	337 (36.1)
Left colon	4016 (25.5)	2478 (21.9)	1028 (24.0)	800 (21.3)	477 (20.4)	173 (18.6)
Rectum	6651 (42.1)	3650 (32.3)	1490 (34.8)	1179 (31.5)	711 (30.4)	270 (29.0)
Other	887 (5.6)	1018 (9.0)	293 (6.8)	333 (8.9)	240 (10.2)	152 (16.3)
**Operative treatment**						
Resection	6878 (43.6)	5146 (45.5)	2071 (48.3)	1816 (48.5)	971 (41.5)	288 (30.9)
No resection	738 (4.7)	314 (2.8)	144 (3.4)	102 (2.7)	54 (2.3)	14 (1.5)
No information/no surgery	8166 (51.7)	5846 (51.7)	2072 (48.3)	1827 (48.8)	1317 (56.2)	630 (67.6)
**Postoperative hospital stay, median in**						
**days (IQR)**						
Colon cancer	6.0 (4.0–8.0)	7.0 (5.0–10.0)	7.0 (5.0–9.0)	7.0 (5.0–10.0)	7.0 (5.0–10.0)	7.0 (5.0–10.0)
Rectum cancer	7.0 (5.0–9.0)	7.0 (5.0–10.0)	7.0 (5.0–10.0)	7.0 (6.0–10.0)	7.0 (5.0–9.5)	7.0 (6.0–9.0)

^1^ FCR, Finnish Cancer Registry classes: 0, unknown; 1, localized; 2, non-localized, regional lymph node metastasis only; 3, metastasized further than to regional lymph nodes or invading adjacent tissues; 4, non-localized, no information on extent; 5, locally advanced, tumor invasion to adjacent tissues; and 6, non-localized, including distant lymph node metastasis.

**Table 2 cancers-16-00135-t002:** Five-year overall survival (%) for colorectal cancer patients according to stage, 2006–2015 (95% CI ^1^).

Age Group	<75 Years Old	75–79 Years Old	80–84 Years Old	85–89 Years Old	≥90 Years Old
FCR ^2^ 0	68.0 (65.8–70.2)	43.9 (39.6–48.2)	37.0 (33.3–40.7)	18.3 (14.6–22.0)	4.8 (2.1–7.5)
FCR 1	88.7 (87.5–89.9)	72.7 (69.4–76.0)	54.6 (50.5–58.7)	38.5 (33.0–44.0)	17.7 (9.7–25.7)
FCR 2	65.6 (63.4–67.8)	52.2 (47.3–57.1)	36.8 (31.7–41.9)	32.3 (25.4–39.2)	14.8 (4.4–25.2)
FCR 3	44.3 (42.5–46.1)	36.7 (33.6–39.8)	26.9 (23.8–30.0)	17.2 (13.9–20.5)	7.4 (3.3–11.5)
FCR 4	52.3 (48.4–56.2)	39.1 (31.7–46.5)	29.8 (22.7–36.9)	19.8 (12.7–26.9)	9.5 (2.1–16.9)
FCR 5	79.1 (77.1–81.1)	60.4 (55.1–65.7)	52.9 (47.2–58.6)	42.8 (35.5–50.1)	19.4 (9.4–29.4)
FCR 6	16.8 (14.6–19.0)	4.1 (1.4–6.8)	4.6 (1.5–7.7)	0.8 (0.0–2.4)	2.1 (0.0–6.2)
All patients	61.7 (60.9–62.5)	47.6 (46.0–49.2)	36.6 (34.8–38.4)	24.5 (22.5–26.5)	9.0 (6.8–11.2)

^1^ CI, confidence interval. ^2^ FCR, Finnish Cancer Registry classes: 0, unknown; 1, localized; 2, non-localized, regional lymph node metastasis only; 3, metastasized further than to regional lymph nodes or invading adjacent tissues; 4, non-localized, no information on extent; 5, locally advanced, tumor invasion to adjacent tissues; and 6, non-localized, including distant lymph node metastasis.

**Table 3 cancers-16-00135-t003:** Cox regression survival analysis by age groups compared with patients <75 years old, hazard ratio (95% CI ^1^), 2006–2015 (*p* < 0.001).

Age Group	<75 Years Old	75–79 Years Old	80–84 Years Old	85–89 Years Old	≥90 Years Old
FCR ^2^ 0	1.00	2.16 (1.91–2.44)	2.83 (2.52–3.18)	4.54 (4.05–5.10)	6.77 (5.93–7.72)
FCR 1	1.00	2.83 (2.43–3.29)	5.10 (4.43–5.89)	7.69 (6.56–9.02)	16.16 (12.88–20.28)
FCR 2	1.00	1.64 (1.43–1.88)	2.31 (2.02–2.65)	3.08 (2.61–3.65)	5.12 (3.94–6.64)
FCR 3	1.00	1.32 (1.20–1.44)	1.80 (1.64–1.97)	2.44 (2.19–2.72)	3.62 (3.06–4.29)
FCR 4	1.00	1.57 (1.32–1.87)	2.03 (1.71–2.41)	2.85 (2.35–3.46)	4.95 (3.82–6.42)
FCR 5	1.00	2.04 (1.73–2.40)	2.99 (2.54–3.51)	4.18 (3.49–5.01)	6.71 (5.15–8.73)
FCR 6	1.00	1.65 (1.46–1.87)	1.96 (1.71–2.25)	3.06 (2.57–3.64)	4.82 (3.58–6.48)
All patients	1.00	1.62 (1.54–1.69)	2.23 (2.13–2.34)	3.21 (3.04–3.38)	5.28 (4.91–5.69)

^1^ CI, confidence interval. ^2^ FCR, Finnish Cancer Registry classes: 0, unknown; 1, localized; 2, non-localized, regional lymph node metastasis only; 3, metastasized further than to regional lymph nodes or invading adjacent tissues; 4, non-localized, no information on extent; 5, locally advanced, tumor invasion to adjacent tissues; and 6, non-localized, including distant lymph node metastasis.

**Table 4 cancers-16-00135-t004:** (**A**) Five-year overall survival (%) for patients with right-sided colon cancer, 2006–2015 (95% CI ^1^). (**B**) Five-year overall survival (%) for patients with left-sided colon cancer, 2006–2015 (95% CI ^1^). (**C**) Five-year overall survival (%) for patients with rectal cancer, 2006–2015 (95% CI ^1^).

Age Group	<75 Years Old	75–79 Years Old	80–84 Years Old	85–89 Years Old	≥90 Years Old
**A**					
FCR ^2^ 0	70.1 (65.6–74.6)	51.8 (43.8–59.8)	43.4 (36.0–50.8)	24.1 (16.5–31.7)	16.8 (7.2–26.4)
FCR 1	87.4 (84.7–90.1)	73.3 (67.2–79.4)	55.4 (48.5–62.3)	41.0 (32.0–50.0)	18.9 (4.2–33.6)
FCR 2	57.7 (53.6–61.8)	49.9 (42.3–57.5)	37.3 (29.3–45.3)	37.5 (27.5–47.5)	20.9 (6.6–35.2)
FCR 3	47.2 (43.9–50.5)	39.3 (34.3–44.4)	29.7 (24.8–34.6)	23.2 (17.1–29.3)	8.5 (2.0–15.0)
FCR 4	47.6 (40.7–54.5)	33.9 (20.8–47.0)	27.5 (16.1–38.9)	24.1 (13.3–34.9)	7.9 (0.0–18.3)
FCR 5	79.9 (76.2–83.6)	64.2 (55.6–72.8)	59.2 (51.2–67.2)	44.0 (32.8–55.2)	17.9 (5.2–30.6)
FCR 6	12.2 (8.9–15.5)	1.6 (0.0–4.7)	5.6 (0.3–10.9)	0.0	0.0
All patients	58.8 (57.2–60.4)	49.2 (46.3–52.1)	39.9 (37.2–42.6)	29.1 (25.8–32.4)	13.1 (8.8–17.4)
**B**					
FCR ^2^ 0	67.4 (62.5–72.3)	40.2 (30.0–50.4)	44.8 (36.2–53.4)	17.4 (9.2–25.6)	0.0
FCR 1	89.7 (87.2–92.2)	73.4 (66.3–80.5)	51.7 (42.7–60.7)	45.1 (32.6–57.6)	32.0 (12.0–52.0)
FCR 2	68.2 (64.1–72.3)	53.0 (43.6–62.4)	37.8 (27.4–48.2)	15.5 (2.4–28.6)	11.8 (0.0–31.0)
FCR 3	43.9 (40.4–47.4)	36.4 (30.3–42.5)	27.0 (20.5–33.5)	13.4 (7.3–19.5)	9.7 (0.0–19.9)
FCR 4	51.7 (43.5–59.9)	33.5 (11.9–55.1)	29.6 (14.1–45.1)	21.4 (6.7–36.1)	0.0
FCR 5	81.9 (78.6–85.2)	60.4 (50.8–70.0)	47.0 (35.0–59.0)	38.2 (24.9–51.5)	13.4 (0.0–35.2)
FCR 6	18.5 (14.2–22.8)	7.0 (0.5–13.5)	0.0	0.0	7.7 (0.0–22.2)
All patients	62.2 (60.6–63.8)	47.6 (44.3–50.9)	36.6 (32.9–40.3)	22.5 (18.4–26.6)	10.6 (5.1–16.1)
**C**					
FCR ^2^ 0	67.3 (64.0–70.6)	38.4 (31.5–45.3)	27.6 (20.9–34.3)	16.4 (10.5–22.3)	1.9 (0.0–4.4)
FCR 1	89.1 (87.3–90.9)	70.3 (65.0–75.6)	55.9 (49.0–62.8)	32.0 (23.6–40.4)	7.5 (0.0–17.1)
FCR 2	70.1 (67.0–73.2)	54.1 (45.9–62.3)	35.9 (27.1–44.7)	34.5 (22.5–46.5)	0.0
FCR 3	44.3 (41.4–47.2)	35.4 (29.9–40.9)	25.8 (20.5–31.1)	15.2 (8.9–21.5)	3.3 (0.0–9.8)
FCR 4	55.4 (48.9–61.9)	45.5 (33.5–57.5)	40.7 (26.6–54.8)	9.0 (0.0–23.3)	5.0 (0.0–14.6)
FCR 5	77.2 (74.1–80.3)	55.0 (45.6–64.4)	47.6 (37.0–58.2)	43.4 (28.7–58.1)	28.8 (2.7–54.9)
FCR 6	19.7 (16.2–23.2)	4.6 (0.0–9.5)	6.0 (0.5–11.5)	0.0	0.0
All patients	64.3 (63.1–65.5)	46.7 (44.0–49.4)	34.7 (31.6–37.8)	22.5 (19.0–26.0)	5.0 (1.9–8.1)

^1^ CI, confidence interval. ^2^ FCR, Finnish Cancer Registry classes: 0, unknown; 1, localized; 2, non-localized, regional lymph node metastasis only; 3, metastasized further than to regional lymph nodes or invading adjacent tissues; 4, non-localized, no information on extent; 5, locally advanced, tumor invasion to adjacent tissues; and 6, non-localized, including distant lymph node metastasis.

**Table 5 cancers-16-00135-t005:** Thirty-day overall survival (%) postoperatively, 2006–2015 (95% CI ^1^).

Age Group	<75 Years Old	75–79 Years Old	80–84 Years Old	85–89 Years Old	≥90 Years Old
FCR ^2^ 0	97.8 (96.8–98.8)	96.6 (94.4–98.8)	93.1 (90.2–96.0)	93.9 (90.2–97.6)	89.4 (80.6–98.2)
FCR 1	99.4 (99.0–99.8)	97.2 (95.6–98.8)	93.4 (90.7–96.1)	93.1 (89.4–96.8)	85.0 (74.0–96.0)
FCR 2	98.9 (98.3–99.5)	97.5 (95.9–99.1)	94.5 (92.0–97.0)	93.4 (89.5–97.3)	95.7 (90.0–100.0)
FCR 3	97.4 (96.6–98.2)	94.4 (92.4–96.4)	91.5 (88.8–94.2)	85.8 (81.1–90.5)	81.8 (72.6–91.0)
FCR 4	97.9 (96.7–99.1)	93.9 (90.4–97.4)	91.3 (87.2–95.4)	87.2 (80.5–93.9)	85.2 (71.9–98.5)
FCR 5	98.5 (97.9–99.1)	96.9 (95.1–98.7)	95.7 (93.5–97.9)	90.4 (86.3–94.5)	90.3 (82.9–97.7)
All patients	98.4 (98.0–98.8)	96.1 (95.3–96.9)	93.2 (92.0–94.4)	90.6 (88.8–92.4)	87.9 (84.2–91.6)

^1^ CI, confidence interval. ^2^ FCR, Finnish Cancer Registry classes: 0, unknown; 1, localized; 2, non-localized, regional lymph node metastasis only; 3, metastasized further than to regional lymph nodes or invading adjacent tissues; 4, non-localized, no information on extent; 5, locally advanced, tumor invasion to adjacent tissues; and 6, non-localized, including distant lymph node metastasis.

**Table 6 cancers-16-00135-t006:** One-year overall survival (%) postoperatively, 2006–2015 (95% CI ^1^).

Age Group	<75 Years Old	75–79 Years Old	80–84 Years Old	85–89 Years Old	≥90 Years Old
FCR ^2^ 0	94.6 (93.0–96.2)	89.5 (86.0–93.0)	84.4 (80.3–88.5)	79.7 (73.6–85.8)	70.2 (57.1–83.3)
FCR 1	97.9 (97.1–98.7)	92.9 (90.4–95.4)	88.0 (84.5–91.5)	84.9 (79.6–90.2)	72.5 (58.6–86.4)
FCR 2	91.3 (89.9–92.7)	89.3 (86.2–92.4)	81.4 (76.9–85.9)	76.1 (69.2–83.0)	61.7 (47.8–75.6)
FCR 3	82.5 (80.5–84.5)	75.0 (71.3–78.7)	68.2 (63.7–72.7)	59.3 (52.6–66.0)	51.5 (39.3–63.7)
FCR 4	88.2 (85.7–90.7)	80.1 (74.2–86.0)	75.4 (69.1–81.7)	70.2 (61.0–79.4)	51.9 (33.1–70.7)
FCR 5	95.9 (94.9–96.9)	92.6 (89.9–95.3)	86.5 (82.8–90.2)	80.9 (75.2–86.6)	69.4 (57.8–81.0)
All patients	91.7 (91.1–92.3)	86.3 (84.7–87.9)	80.5 (78.7–82.3)	75.1 (72.4–77.8)	62.9 (57.4–68.4)

^1^ CI, confidence interval. ^2^ FCR, Finnish Cancer Registry classes: 0, unknown; 1, localized; 2, non-localized, regional lymph node metastasis only; 3, metastasized further than to regional lymph nodes or invading adjacent tissues; 4, non-localized, no information on extent; 5, locally advanced, tumor invasion to adjacent tissues; and 6, non-localized, including distant lymph node metastasis.

## Data Availability

Given the large series of datasets from FCR and patient confidentiality, we are not at liberty to release the data in its current form.

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
