# Peer review of "Short- and Long-Term Survival among Elderly Colorectal Cancer Patients in Finland, 2006–2015: A Nationwide Population-Based Registry Study"

_cancers, 2023, doi:10.3390/cancers16010135_

Round 1
Reviewer 1 Report
Comments and Suggestions for Authors
I found this article to be a rational and solid investigation of colorectal cancer survival across the population of Finland. It examined several survival outcomes (post-operative, disease-specific, short-term and long-term) in older people (those ≥75 years compared to <75 years) and made comparisons by both tumour site and stage of disease. As such, the paper contains a lot of information that was succinctly and well presented. However, as neatly explained in the introduction, the challenge for those treating the disease is that the population is ageing, so incidence is increasing and the appropriate management of older people, when they may possess other conditions and a limited life expectancy, is difficult. Whilst the figures in this paper present a clear descriptive analysis of what survival rates could be expected I do feel the analyses could go further to help readers understand why survival varies by age and this, in turn, would help better inform care and decision making in older people.
For example, all the analyses presented appear to use observed survival. Cancer registry survival analyses frequently use relative- or net-survival. This method has the advantage of quantifying the excess mortality bought about by the cancer compared to the survival of the population as a whole. Incorporating such an analysis would allow quantification of what survival outcome could be expected at a given age if the cancer was, or was not, treated. Such information would, perhaps help those diagnosed with the disease, and those treating them, to make informed decisions about their management strategies.
The study uses the under 75s as a comparator population. Such a group will contain a lot of people of many different age groups who may all have different appetites for treatment and also different long-term outcomes. Would it be sensible to expand the resolution of the age groups? For example, ≤60, 60-65, 66-70, 71-5?
Throughout the paper, a number of different survival outcomes are reported across lots of age groups as well as by tumour site and stage of disease. As presented in the paper, the distribution of such characteristics vary by age (Table 1). For example, older people with colorectal cancer are more likely to be female, have right-sided disease, present as an emergency, have unreported stage and less likely to have surgery (at least in the over 85s). How does this variation in patient characteristics influence management and, in turn, the different survival outcomes observed? Greater discussion of this would help move the paper from being purely descriptive to providing stronger translational evidence.
The Finnish Cancer Registry (FCR) staging classification is used and the paper states this is broadly similar to TNM classification. Although I am familiar with the TNM staging system I wasn’t overly clear of the relationship. Additional information to provide greater clarity on this would be helpful.
The number of patients for whom operative treatment information is captured is low with information being lacking (or surgery not happening) in around 51.7% of cases. This is quite a substantial absence of data so what implications does this have on how the post-surgery survival outcomes should be interpreted.
Post-operative hospital stay is included in Table 1 but the relevance of this information isn’t really highlighted. What is the importance of such information and how should it be interpreted alongside the survival outcomes.
Finally, in paragraph 2 of the discussion the authors clearly describe the challenge. Older people have lower life expectancy those younger people irrespective of whether then have cancer or not and so seeing lower survival is perhaps not unexpected. However, age is just a number and some older people may be fitter than their younger counterparts. As such, nsuring the optimal treatment strategy should not be based on age alone. Rather than just describe this, however, I feel the paper could be substantially improved by better analysing the data to try and quantify any inequalities in care and outcome and what may be driving them. In this way the data could be used to try and inform strategies to tackle true age-related inequities in colorectal cancer outcome in Finland.
Reviewer 2 Report
Comments and Suggestions for Authors
The authors describe in a large Finnish cohrot the epidemiology and prognosis of CRC in the elderly. The data is quite extensive and of high interest for the readers of cancer.
Please give a tabel where the FCR and the UICC staging is compared so the reader better understands the staging used in this article.
Please discuss the relatively high percentage of FCR0 patientn in the very elderly. Is ther a less rigorous staging appiliee? What are the reasons? Given the dtata on the younger old age group, the question is whether there is a diagnostic nihilism in the elderly population which may in part explain the worse outcome of this group. Also give the median life expectancy for 90+ old patients to understand the natural death rate in these patient group.
Round 2
Reviewer 1 Report
Comments and Suggestions for Authors
The authors of the paper have responded adequately to all the comments I made. Further detail would be appreciated, for example, why absolute survival is preferable to relative survival. They state absolute survival is more valuable to 'clinical work' but not why and I would suggest that more novel use of the data to throughout would increase clinical relevance. However, the responses are adequate and acceptable.
